# Nitrification Process in a Nuclear Wastewater with High Load of Nitrogen, Uranium and Organic Matter under ORP Controlled

**Mariano Venturini** [1,*], **Ariana Rossen** [2] and **Patricia Silva Paulo** [1]

1 Biomining and Environmental Biotechnology Laboratory-National Commission of Atomic Energy, Av. Ptero González y Aragón N°15, Buenos Aires B1802AYA, Argentina; silvapau@cae.cnea.gov.ar
2 Microbiology Laboratory, Water Use Technology Center, National Water Institute, Au, Ezeiza Cañuelas, Tramo J. Newbery km 1,6., Buenos Aires B1804, Argentina; arossen@ina.gob.ar
* Correspondence: mventurini@cnea.gob.ar; Tel.: +54-11-4125-8531

**Abstract:** To produce nuclear fuels, it is necessary to convert uranium's ore into $UO_2$-ceramic grade, using several quantities of kerosene, methanol, nitric acid, ammonia, and, in low level, tributyl phosphate (TBP). Thus, the effluent generated by nuclear industries is one of the most toxic since it contains high concentrations of dangerous compounds. This paper explores biological parameters on real nuclear wastewater by the Monod model in an ORP controlled predicting the specific ammonia oxidation. Thermodynamic parameters were established using the Nernst equation to monitor Oxiders/Reductors relationship to obtain a correlation of these parameters to controlling and monitoring; that would allow technical operators to have better control of the nitrification process. The real nuclear effluent is formed by a mixture of two different lines of discharges, one composed of a high load of nitrogen, around 11,000 mg/L ($N-NH_4^+-N-NO_3^-$) and 600 mg/L Uranium, a second one, proceeds from uranium purification, containing TBP and COD that have to be removed. Bioprocesses were operated on real wastewater samples over 120 days under controlled ORP, as described by Nernst equations, which proved to be a robust tool to operate nitrification for larger periods with a very high load of nitrogen, uranium, and COD.

**Keywords:** thermodynamic parameters; nitrifying bacteria; bioprocess; nuclear effluent

## 1. Introduction

Nowadays, different societies around the world are claiming for more formal and robust legislation focused on environmental protection. Along with the development of the nuclear industry and its prospective applications on a large scale [1], there is an increased radioactive effluent that requires a stricter environmental normative. The International Atomic Energy Agency (IAEA) and Argentine legislation contemplate that effluent discharges from nuclear industries, considered as radioactive waste, should be zero; even though radioactive emissions remain at a harmless level [2].

Other industries such as explosives or exchange resin have effluents containing high levels of nitrogen that can be used as fertilizers [3,4]. However, effluents from the nuclear industry cannot further be used because they are considered hazardous for the environment and human health [5], and Argentinean legislations prohibit further uses of nuclear effluents [6].

If radioactive effluents are discharged directly into the environment, they will pollute water, atmosphere, and soil, threatening the environment and human health [7]. So, radioactive effluent treatments are becoming one of the most important issues that environmental legislation has to address [8].

To produce nuclear fuels, it is necessary to convert uranium's ore into $UO_2$-ceramic grade, using several quantities of kerosene, methanol, nitric acid, ammonia, and, at a low level, tributylphosphate (TBP) [9]. Thus, the effluent generated by nuclear industries is one of the most toxic since it contains high concentrations of toxic compounds.

In general, physicochemical treatments such as ammonia stripping, reverse osmosis [10,11], or Zero Liquid Discharge (ZLD) [12] are cost-effective methods employed for highly concentrated wastewater containing more than 1000 mg N/L of the ammonium ion despite its high costs [13]. Other physicochemical strategies implemented are coupled to ionic exchange with reverse osmosis [10,14,15] to obtain a better performance, though it requires recycled resin generating a secondary effluent. However, the cost-effectiveness of those physicochemical treatments increases dramatically.

By contrast, the biological nitrification and denitrification process seemed to be a promising technology to remediate nuclear effluent [16–18]. However, due to the high quality and quantity of nuclear effluents variability, biological treatments are difficult to implement with real samples [19]. Besides, microbial kinetic performance is lower than physicochemical ones, and even more, considering the complexity of interactions it is difficult to estimate the evolution of the microbiological process [20].

Wastewater nitrogen removal rates depend, in part, on the complete oxidation of ammonia [21]. Complete nitrification is carried out by two different microbial groups that can be studied separately as: the ammonia-oxidizing bacteria (AOB) that converts ammonia into nitrite, and then, the nitrite-oxidizing bacteria (NOB) that convert nitrite into nitrate, in a process called nitritation [22].

The last innovative technologies [23,24] are System for High-Activity Ammonium Removal Over Nitrite [SHARON], Anaerobic Ammonium Oxidation (ANAMMOX) [25,26], Complete autotrophic Nitrogen Removal Over Nitrite (CANON) [27,28] and Simultaneous Nitrification-Denitrification (SND) [29]. Although this process is difficult to implement in an effluent with a high load of COD and uranium due to its toxicity levels. So, in general, tertiary treatment of waste is needed [30,31].

To monitor the switch "on and off" for each aeration period, oxidation-reduction potential (ORP) and pH are the key parameters to control the nitrification process [32]. Because ORP in the anoxic phase provides indication [24,33] for total nitrogen removal efficiency and represents a quantitative indicator of the degree of completion of the mentioned chemical reaction [34,35]. So, ORP becomes a valuable parameter for real-time online control of nitrification and can be easily monitored using a peak signal on the controlling screen indicating the final of the nitrification process.

According to Chang [36], Nernst equations for online control of the biological process were based on a one-to-one stoichiometric relation for the oxidizing and the reducing species. In the present work, ORP changes in the treatment of nuclear wastewater effluent were investigated.

To do that, the nitrification process of a real wastewater sample was modeled and simulated studying ORP by thermodynamic equations to obtain parameters for optimum biological process in complex media at real conditions. The biological kinetics were studied by the Monod equation and the continuous DO parameters by online methodologies [37].

## 2. Materials and Methods

### 2.1. Isolation Synthetic Medium

Due to extreme physicochemical conditions of nuclear wastewater, no nitrification activity was detected in those effluents. Therefore, nitrifying biomass was isolated from a soil sample in a synthetic medium, and then acclimated to an organic load and real and nuclear wastewater.

The autotrophic medium was described by Kassen [38], per liter, $SO_4(NH_4)_2$ 1 g, 6 g of $CaCO_3$, 2 g of $NaCl$, 0.2 g of $MgSO_4 \cdot 7H_2O$, 0.5 g of $KH_2PO_4$, and 1 mL of traces elements 1 g Fe III-Citrate, 10 g of $MnCl_2 \cdot 4H_2O$, 10 g of $H_3BO_3$, 5 g $ZnCl_2$, 1 g of $Na_2MoO_4 \cdot 2H_2O$, 4 g of $LiCl$, 2.5 g of $KBr$, 2.5 g of $KI$, 5 g of $CoCl_2$, 0.5 g $SnCl_2 \cdot H_2O$, 1 g of $AlCl_3$, 20 g of $EDTA$, and 0.05 g of $CuSO_4$. The inoculum was initialized at 5% *v/v* at 25 °C, 120 rpm in a 500 mL Erlenmeyer flask with a final volume of 250 mL. Isolation of nitrifying bacteria

were enriched and fed using ammonia as the energy source and carbonate as a carbon source in function of pH and the following parameters equation

$$H_2O + CO_2 \Leftrightarrow H^+ + HCO_3^- \Leftrightarrow H_2CO_3 \text{ with } K = [H^+] \cdot [HCO_3^-] / [H_2CO_3].$$

### 2.2. Blended Real Nuclear Wastewater (BRNW)

Fuels from the nuclear power plant are obtained by uranium ore's processing which begins with the leaching of uranium ores converted into a "yellow cake" ($U_3O_8$) that produces uranyl nitrate solution (200 g U/L) by dissolution in nitric acid [39,40]. These solutions carry out the precipitation of uranium oxide with anhydrous ammonia to produce ammonium diuranate (ADU).

Real nuclear wastewater samples consisted of mixing effluent streams from the argentine uranium conversion facility. This Blended Effluent from Real Nuclear Wastewater, named as BRNW hereafter, was composed of a mix of different discharge flows, to neutralize and dilute nitrogen concentration [41].

The first (1) stream was the current processing characterized by a pH = 1 and a concentration of nitrogen that varies between 11,000–14,000 mg/L (3 N-NH$_4^+$/1 N-NO$_3^-$). The second (2) stream came from a domestic industrial washing equipment effluent with a pH = 9 and 2700 mg/L total organic nitrogen and 13,000 mg/L of COD with small quantities of TBP and detergent. Both effluents were mixed in such a relationship as to ensure that the final parameters were: COD 1500 mg O$_2$/L, N-NO$_3^-$ 1000–1400 mg/L, and N-NH$_4^+$ 600–1400 mg/L (around 1/10). The selected values of the parameters were obtained from the Monod maximum velocity (Vmax) and ORP determination, with the final pH adjusted to pH = 7.

Table 1 reveals that there have been three different periods to make acclimation. Strategies were carried out in three stages for 40 days. Stage 1, a synthetic media without nitrogen source, received a volume from stream 1 sample up to obtain a 500–700 mg/L ammonium concentration in the media. Stage 2, was a synthetic medium with a sample of stream 2, incorporating COD to produce heterotrophic acclimatation. The final step consisted of an adaptation of the culture to BRNW reaching a pH, nitrogen content, and COD to optimal growth of nitrifying bacteria.

**Table 1.** Characteristics of the acclimatation media for bacterial adaptation.

| Parameters | Unit | Acclimation-Autothropic Medium-Nuclear Effluent (Stage 1) Concentration Minimum-Maximum | Acclimation-Heterothropic Medium (Stage 2) Concentration Minimum-Maximum | Blended Real Nuclear Wastewater (BRNW) Concentration Minimum-Maximum |
|---|---|---|---|---|
| Ammonium | mg/L | 500–700 | 600–900 | 600–1400 |
| Nitrate | mg/L | 0–400 | 100–800 | 1000–1300 |
| COD | mg/L | 0 | 9000 | 1500 |
| pH | | 6.0–8.0 | 6.0–8.0 | 7.0–7.8 |
| Uranium | mg/L | 300–600 | 0 | 300–600 |

### 2.3. Test Analysis

The system was monitored using chemical and physical analyses of the influent and effluent based on the procedures described by Standard Methods for the Examination of Water and Wastewater [42]. Analyzed parameters were: pH (4500-H B); chemical Oxygen Demand (COD-5220 D); nitrite N-NO$_2^-$ (4500 B-FIA); nitrate- N-NO$_3^-$ (4500 A). An analysis of the NH$_4^+$ concentration was conducted using spectrophotometric determination by phenate method at 640 nm (4500-NH$_3$ FB), NO$_2^-$ (4500 B-FIA) was completed using diazotization methods and 220 nm and Nitrate by absorption at 220/270 nm (4500 A-FIA).

### 2.4. Monod Kinetics Model

The specific nitrification rate is commonly expressed by the multiplication of individual terms complexing the Monod-type expressions [43]. The used model was carried out with 10% of inoculum at 25 °C and 120 rpm, to maintain pH = 8, supplemented with insoluble $CaCO_3$.

Based on the above considerations, the following expression for the growth rate of ammonium oxidation were determined by linearization of Lineweaver-Burk, where Vmax and the ammonia affinity constant ($K_{NH4}$) were obtained under controlled incubation conditions in the Erlenmeyer flask [44]. The maximum nitrification rate (Vmax) was obtained by measuring initial velocity consumption of $N-NH_4^+$ mg/L day in a batch reactor operated at $25 \pm 1$ °C. Non-limiting DO conditions ($[O_2] > 5$ mg/L), initial concentration at 480 up to 2300 mg/L [$N-NH_4^+$] were used. The inhibition of $K_i$ and $HNO_2$ concentration was adjusted by the Haldane model as described by Carrera Muyo [45].

$$V = V_{max}\left[\frac{S_{(N-NH_4^+)}}{(K_s + S)\cdot\left(1 + \frac{HNO_2}{K_i}\right)}\right] \tag{1}$$

where: V = nitrification rate ($N-NH_4^+$ mg/L·day), Vmax = Maximum nitrification rate ($N-NH_4^+$ mg/L·day), S = Ammonium Substrate Concentration (mg/L), Ks = Half Saturation (mg/L), $HNO_2$ = nitrous acid concentration ($N-NO_2^+$ mg/L), Ki =Inhibition constant by $HNO_2$ (mg/L).

Linear respirometry was verified by using $Na_2CO_3$ 1M consumed accordingly with the nitrogen ammonia concentration. Respirometry assays were performed according to Weissman-Ciudad methodologies [46] [47]. During online determination, linearization was considered between the range of 4 and 6.5 mg $O_2$/l. The respirogram was carried out in Sartorius 5l bioreactors where DO and pH were controlled by Mettler Toledo Sensor. The equation used is as follow:

$$(NH_{4(i)})/Ln\frac{NH_{4(I)}}{NH_{4(f)}} = (R_{AOB}\cdot T - K_s\,(AOB))/(Ln\frac{NH_{4(I)}}{NH_{4(f)}}) \tag{2}$$

where $NH_{(48i)}$ = Initial concentration of ammonium $N-NH_4^+$ mg/L, $NH_{(48f)}$ = final concentration of ammonium $N-NH_4^+$ mg/L, $R_{AOB}$ = AOB DO rate mg/L.hour $K_s$ (AOB) = Oxygen saturation constant and T = time (hour)

The Monod nitrification rate was:

$$V = 4.55\cdot V_{max}\cdot(7.2 - pH)\cdot e^{\theta(T-Tr)}\cdot\frac{(DO)}{K_{pO_2} + DO}\cdot\left(\frac{(N-NH_4^+)}{(K_{NH_4^+} + N - NH_4^+)\cdot(1 + (\frac{HNO_2}{K_i}))}\right) \tag{3}$$

where V ($N-NH_4^+$ mg/L·day) and Vmax ($N-NH_4^+$ mg/L·day) are Monod parameters, DO = dissolved oxygen (mg/L), $HNO_2$ = Nitrous acid concentration (mg/L), T = temp and $T_r$ = ref. temp. (°C) and $K_1$ = inhibition constant by $HNO_2$ (mg/L). Operational factor 4.55 [Y(x/s)] was incorporated to obtain a better performance according to empirical data.

Inhibition was considered as "competitive inhibition" based on the Haldane model. The linearization was operated with a low concentration of $HNO_2$. The method used in this work to determine nitrification inhibition was the Haldane model. According to Carrera Muyo [45] S>>>KS, consequently, the equation was as follows:

$$\frac{1}{V} = \frac{1\cdot HNO_2}{V_{max}} + \frac{1}{V_{max}} \tag{4}$$

V($N-NH_4^+$ mg/L·day) and Vmax($N-NH_4^+$ mg/L·day) are Monod parameters, and $HNO_2$ Nitrous acid concentration(mg/L).

The influence of temperature on biological activity is often modeled by an Arrhenius equation: $V = V_{max}\cdot e^{\theta(T-Tr)}$ where $V_{max}$ (T) is the maximum specific growth rate and V

is the growth rate at the actual temperature T-Tr is the reference temperature and θ is the Arrhenius constant.

The theoretical reaction shows that approximately 1mol of alkalinity (as $Na_2CO_3$) is consumed for every mol of ammonium oxidized [48].

$$FA = TAN \cdot \left( \frac{1}{10^{(pH-pKa+1)}} \right) \rightarrow pKa = 0.09 + \frac{2730}{T\ (K)} \tag{5}$$

where FA = Free Ammonia (mg/L), TAN = Total Ammonia Nitrogen (mg/L), pKa = Affinity constant ($NH_4^+$) and T = temperature (K°)

Proton production was proportional to oxidized ammonium, and to maintain pH, equivalent volume (ml) of carbonate solution were pumped into the bioreactor. Therefore, TAN could be estimated during online processes.

$$TAN = TAN - \left( \frac{ml \cdot Vs\ 14gl\frac{N}{mol}}{Y_{\frac{CO_3^{2-}}{N-NH_4^+}} \cdot TAN} \right) \tag{6}$$

TAN = Total Ammonia Nitrogen (mg/L), N = nitrogen, $Y_{\frac{CO_3^{2-}}{N-NH_4^+}}$ = Carbonate/ammoniun yield, and ml·Vs = ml Carbonate added.

To corroborate online ammonium oxidation, the consumption of 1 M carbonate was utilized to monitor the process. These studies were evaluated using a "point-to-point" rate (Velf-Vel0)/(final time-initial time).

*2.5. Control of Environmental Conditions to Monitor ORP and in Relation to Monod Parameters*

The ORP and pH were determined by Mettler Toledo sensor pH2100. Ammonium concentration (N-$NH_4^+$), nitrite (N-$NO_2^-$), and nitrate (N-$NO_3^-$) were carried out according to standard methods (See Section 2.3).

The ORP model was proposed by Chang-Cheng [36] based on the Nernst equation

$$E = E° + \frac{RT}{nF} \cdot Ln\left[NH_4^+\right] + \frac{2RT}{nF} \cdot Ln[DO] + \frac{2RT}{nF} \cdot Ln\left[\frac{1}{NO_3^-}\right] + \frac{2RT}{nF} \cdot Ln\left[\frac{1}{H_2O}\right] + \frac{2RT}{nF} \cdot Ln\left[\frac{1}{NO_2^-}\right] + \frac{2RT}{nF} \cdot Ln\left[\frac{1}{H}\right] \tag{7}$$

E = ORP value (mV), E° = Standard Potential (mV), RT/nF Nernst parameters, nitritation and nitratation: DO, $NH_4^+$, $NO_2^-$, $NO_3^-$, (mg/L) and H = proton reduction.

The standard potential (E°) used in this work were ammonium oxidation to nitrite ($NO_2^-$) = 154 mV and nitrate $E^0$ ($NO_3^-$) = −340 mV. The original equation was modified by replacing ln(1/[H+]) for 2.3026 xxx 4 pH according to the author:

$$E = E° + \frac{RT}{NF} \cdot Ln\left[ \frac{(NH_4^+) \cdot (DO) \cdot (H_2O)}{(NO_x^-) \cdot (H)^{(8\ or\ 6)+}} \right] \tag{8}$$

$$E_0\ (NO_2^-) = 154\ mV\ E_0(NO_3^-) = -340\ mV$$

$$E = -340\ mV + \left( \frac{0.059}{6} \cdot Log\ DO \right) + \{61 \cdot pH - 59.88 \cdot Log\ \frac{[NH_4^+]}{[NO_x^-]}\} \tag{9}$$

Dissolved Oxygen (DO) concentration was considered constant due to agitation and forced air injection rate throughout the test and at an initial stage when biomass did not have a considerable consumption of oxygen. This value was 8 mg/L at 25 °C, according to the standardized table ($pO_2$).

*2.6. Inhibition Assays: Maximum Load of Blended Real Wastewater*

To study the toxicity effect of the effluent, inhibition of Nitrifier bacteria was determined by ammonium oxidation rate. Using a pre-established culture in Erlenmeyer flasks, BRNW samples were added from 5% up to 50% (5, 10, 15, 20, and 50%) to the autotrophic medium.

To compare kinetic performance between biofilm vs. planktonic cells removal efficiency was evaluated using hydrogels as a carrier. Biomass was cultivated over 100 days with a semi-continuous process [49]. To do that, the Erlenmeyer flask was inoculated with 50 mL of nitrifying bacteria of approximately 5%$v/v$ per liter with hydrogel carriers.

In the first lag period of the nitrifying process, no organic carbon sources were provided to reduce heterotrophic growth. After that stage, once the biofilm was formed onto hydrogel carriers, BRNW was added at an increasing concentration to adapt biomass.

*2.7. Pre Scale-Up Operations in Moving Bed Bioreactor (MBBR) using BRNW*

The biomass added onto carriers were cultivated in autotrophic media for 40 days in Sartorius Biostat APlus Bioreactor. After that, biomass was acclimated to BRNW for another 40 days. The ORP value was kept above 100 mV to guarantee the nitrification process. To establish the best nitrifier performance and avoid nitrite accumulations, the bioassay was carried out under two different values of pH, 7.2 and 7.8. Obtaining better results at 7.2 without nitrite accumulation. The oxygen was supplied with a constant flow of 2 lpm of air and 200 rpm.

*2.8. Mathematical Models*

Linear and nonlinear regression analyses were used to fit the best baseline data for uptake. Nitrification rates with increasing N-NH$_4^+$ concentrations were analyzed with nonlinear regression using Origin Microcal 8.0: computes, squares estimates were given by nonlinear Hill with n = 1. All linear regression were applied by default method.

## 3. Results and Discussion

*3.1. Monod Kinetic Parameters*

It is commonly assumed that nitrification bioprocess has zero or first-order kinetic, this assumption does not consider the role of biological factors and environmental influence over bacterial metabolism. Monod model is used to describe growth rate and inhibition growth factor, therefore, in this work, it is proposed a Monod model under a high load of nitrogen condition to describe a nitrification rate.

Figure 1A shows kinetic nitrification obtained by the Monod model (Hill->n = 1). The rose circle line represents the negative control (absence of microorganisms). At a higher concentration of ammonium, the lag phase takes longer times. Ammonium curves decline at different times over the incubation period: 7, 10, 12, and 13 days for 240, 490, 900, and, finally, 1380 mg/L N-NH$_4^+$ respectively. It could be explained for the adaptation time to free ammonia concentration [50].

Even though, at a higher concentration, it requires a longer lag phase, the velocity of the reaction is increased as it is shown in Figure 1B as the first derivative. The velocity had an inversely trend at initial ammonium concentration showing a more abrupt decline slope, according to the Monod model in function of the initial substrate concentration. Therefore, for higher concentrations, the velocities of ammonium oxidation were higher as well, which represents an important parameter for the bioprocess of BNRW.

Figure 1C,D show the maximum velocities that fit the Monod equation and the results obtained by linearization (following the Lineweaver–Burk method), respectively.

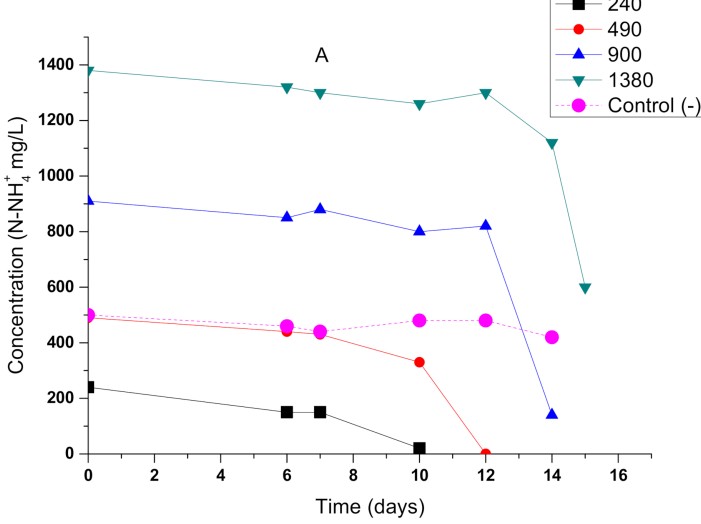

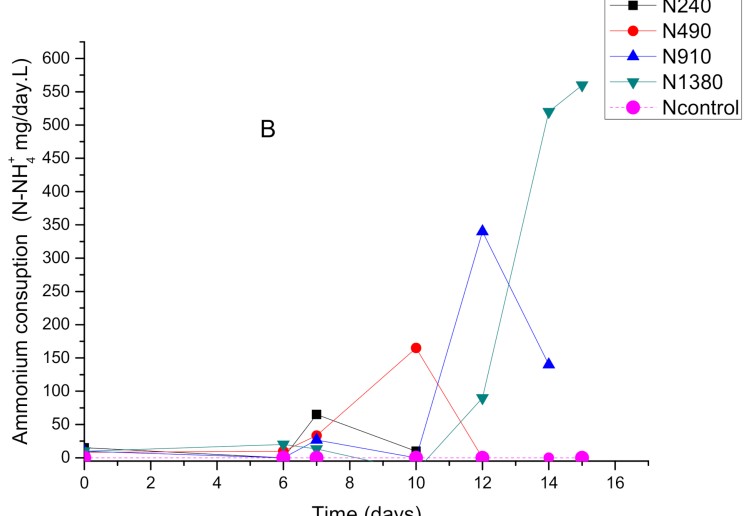

**Figure 1.** *Cont.*

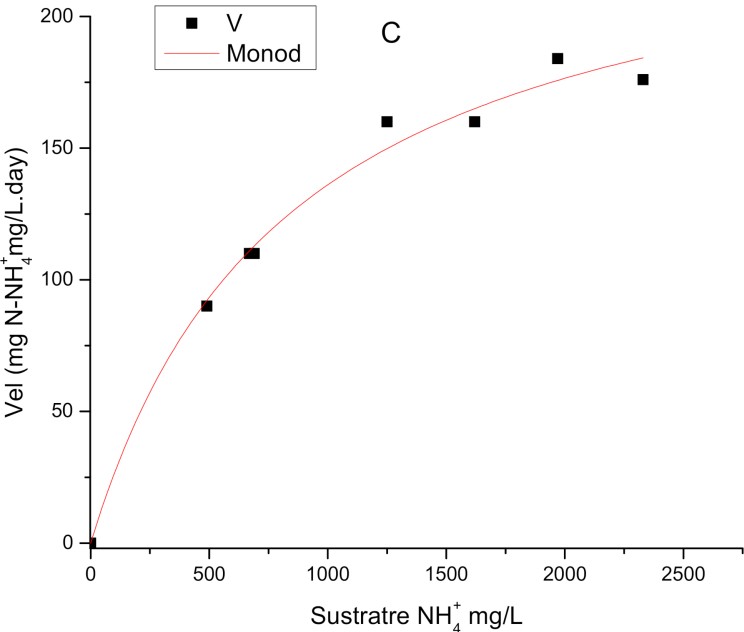

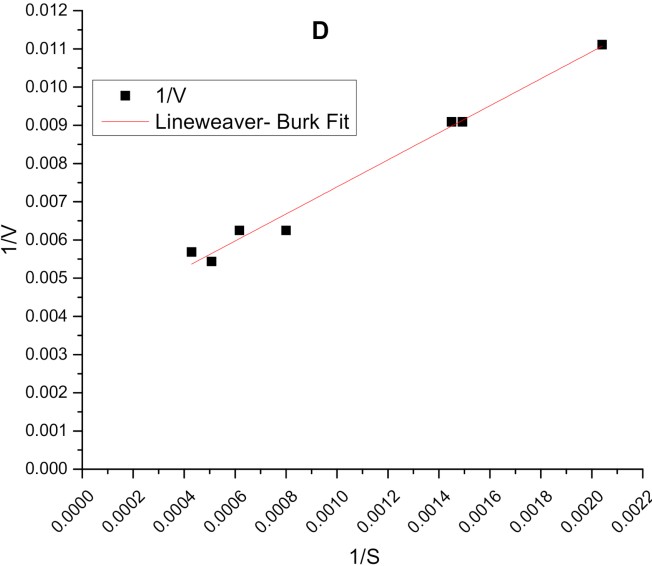

**Figure 1.** (**A**) Effect of initial ammonium concentration during lag phase. (**B**) Velocity evolution over time as a function of substrate addition. (**C**) Maximum velocity as a function of initial substrate and time (**D**) linearization.

Figure 2 shows that the maximum velocity was restricted to 1380 mg/L of ammonium, due to Ammonia/Ammonium equilibrium which also depends on pH and concentration (pKA). Free ammonia ($NH_3$) can easily penetrate the cells producing a competitive inhibition effect, affecting biological activity according to pH and pKa values.

As was demonstrated by Anthonisen [51], uncharged ammonia is a substrate for the ammonium oxidizer bacteria and can easily penetrate the membrane cell. In this work, it was also observed, an inhibition process by substrate and is related to pH by the following expression:

$$S_{NH_3} = \left(\frac{NH_3}{NH_4^+}\right) / \left[1 + \left(\frac{10^{-pH}}{K_{eq}.NH_4^+}\right)\right] \tag{10}$$

Being $S_{NH_3}$ = susbtrate ammonium, $NH_3$ and $NH_4^+$ (mg/L) and $K_{eq}$ = equilibrium constant.

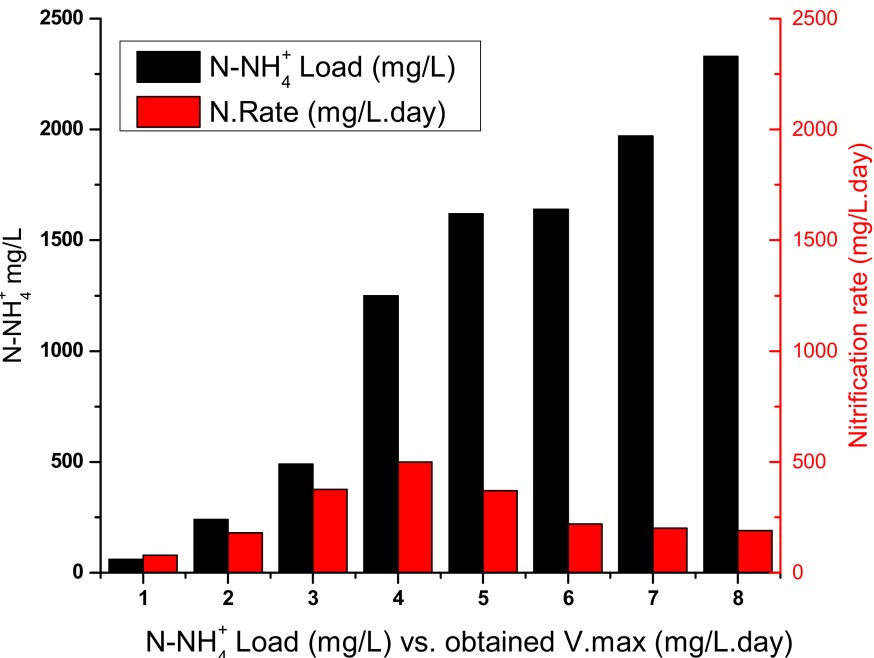

**Figure 2.** Maximum load of $N-NH_4^+$ and nitrification performance.

This expression illustrates the relationship between ammonia/ammonium modified by pH and their concentrations and impacts the kinetics inhibition.

As it was mentioned before, the results of this study indicate that the maximum load of ammonium (1380 mg/L $N-NH_4^+$) reaches the Vmax (251 $N-NH_4^+$ mg/L·day).

Figure 3 shows the linearization used to obtain the inhibition constant by nitrous acid ($HNO_2$) according to the Haldane model. Similarly to free ammonia, the uncharged $HNO_2$ can penetrate the cells considering a pKa of 4.5. As with ammonia, low levels of nitrite should be maintained to avoid inhibition.

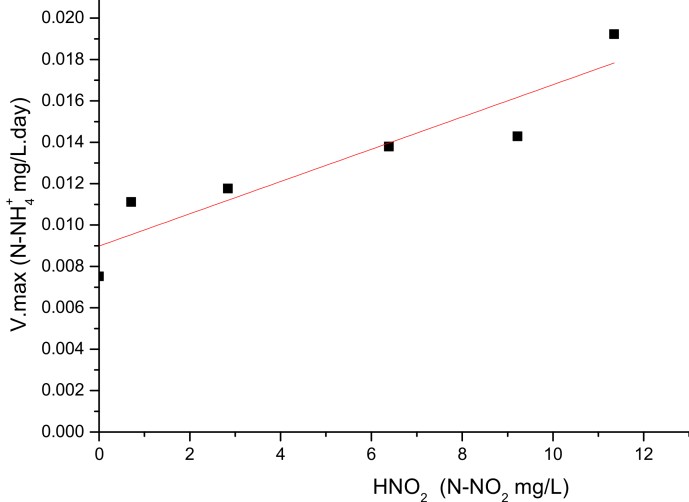

**Figure 3.** Competitive inhibition by $HNO_2$.

Another crucial factor in this type of process is the dissolved oxygen level [52,53]. The oxidation reaction was ensured at a high level of DO supply to the medium to avoid having a limiting factor to autotrophic bacteria. Because of that, oxygen was strictly maintained

between 4 to 6 mg/L (DO) to ensure not only limitations to autotrophic bacteria but also be adjusted to a linear fit according to Ciudad [46,47]. The results obtained by this methodologies are shown in Figure 4.

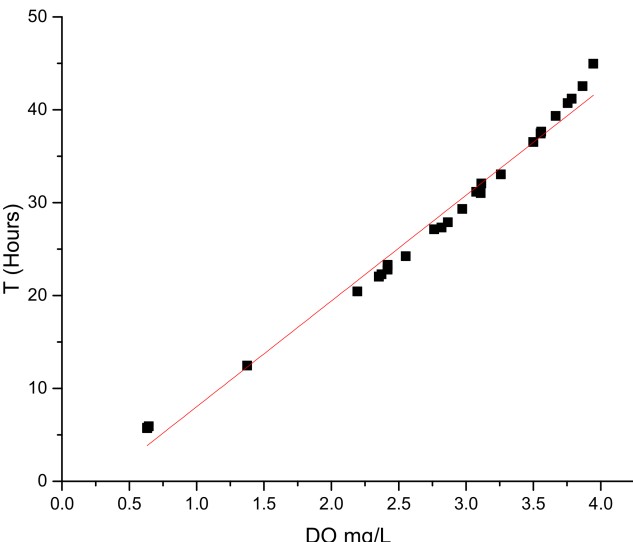

**Figure 4.** Linearization of respirometric kinetic.

The last parameter studied in this work was the proton production by nitrification. In Figure 5 the correlation between oxygen consumption and pumped carbonate was empirically confirmed. Levels of DO were maintained stable up to 120 h coincidently with catalytic activity, after that, the dissolved oxygen decreased progressively. Simultaneously, $Na_2CO_3$ (1 M) was added to the bioreactor at the same time to counterbalance the pH changes. As it is shown in Figure 5, the relationship between DO uptake and the added volume of $Na_2CO_3$, results in complete oxidation of ammonium conversion, similarly to previous studies [37]. These results support the conclusion that these parameters such as DO and carbonate consumption are useful tools for monitoring the nitrification performance.

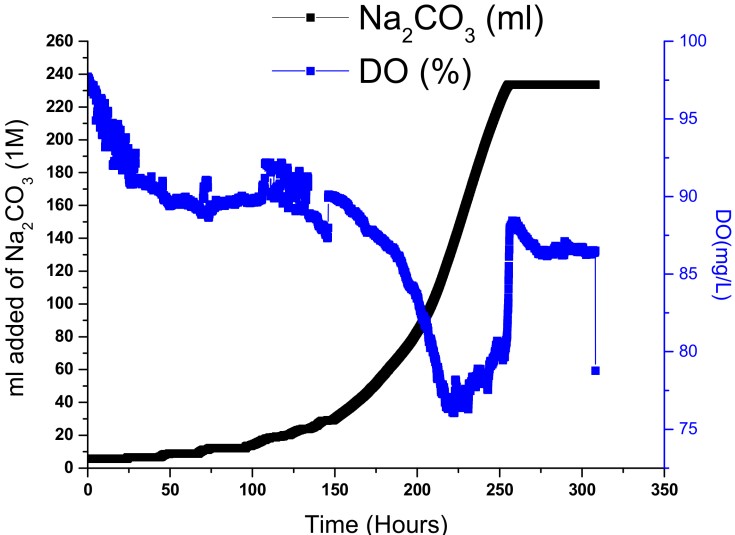

**Figure 5.** Relationship between DO consumption and pumped $Na_2CO_3^{-2}$ 1 M.

As a result, to establish the operational range of the parameters of the nitrification process, a Monod model was used, considering $NH_4^+$ concentration, pH, and temperature as independent variables and DO and $HNO_2$ as dependent variables. The values of the

parameters obtained are as follows: Vmax: 251 mg/L·day $NH_4^+$; DO 4–6 mg/L-Maximum ammonium load: 1380 mg/L $N\text{-}NH_4^+$. In the case of pH, it was adjusted to 7.2, even though it was demonstrated that nitrifying bacteria has an optimal metabolism at pH 7.8 (from experimental data). In this work, it was observed that at that pH nitrite accumulates.

$$V = 4.55 \cdot \left\{ \left( 251 \frac{\text{mg}}{\text{l}} \cdot \text{day} \right) \cdot (7.2 - \text{pH}) \cdot e^{\theta(T-Tr)} \cdot \left[ \frac{DO(4-6 \text{ mg/L})}{(Ks_{OD} + DO)} \right] \cdot \left( \left[ \frac{(N-NH_4^+)}{(N-NH_4^+) \cdot \left(1 + \left(\frac{HNO2}{1.25}\right)\right)} \right] \right) \right\} \quad (11)$$

Being, correction factor = 4.55, T = optimal temperature (°C), Tr = reference temperature (°C), DO = Dissolve Oxygen (mg/L), $KS_{DO}$ Affinity constant of oxygen (mg/L), $HNO_2$ = Nitrous acid concentration (mg/L).

A common problem regarding this model is its application to real effluent conditions that differ from the standardized synthetic medium. When nitrification assay is run using BNRW, heterotrophic biomass already presents competes with autotrophic bacteria due to the high metabolic rate, affecting the nitrification performance.

### 3.2. Control of ORP and Relation with Monod Parameters under Autotrophic Conditions

Nitrification kinetics can be studied under environmental conditions with high alkalinity, COD, and DO levels, using oxide-reduction potential (ORP) as a control parameter to establish nitrification conditions. The parameters used to stimulate autotrophic biomass should maintain an oxidative environment and can be monitored by ORP values.

The biotechnological challenge is to control the efficiency by autotrophic biomass and to prevent unwanted inhibition conditions due to semi-continuous load and wastewater flow. In the bibliography, there are a scarce number of studies that use a dynamic ORP control technique to complement other parameters such as pH and/or DO to monitor nitrification processes. In those studies, the use of the Nernst method was suggested to obtain information related to nitrogen removal dynamics as was applied to the model proposed by Chang [36,54].

To obtain the relationship between ORP with other control parameters as pH and a load of $NH_4^+$ under controlled conditions, several assays were carried out using the synthetic medium. Ammonia is oxidized to $NO_2^-$, as a consequence of a bacterial shift, under aerobic culture based on a one-to-one stoichiometric relation for the oxidizing and the reducing species. Figure 6 shows changes in ORP, pH, $NH_4^+$, and $NO_2^-$ during a ten days treatment assay.

At the initial stage from day 1 to day 4, ORP values increased from 0 mV to 100 mV, when the temperature cooled down from sterilization to room temperature, and DO reach normal levels. During the second stage, from day 4 to 8 of the kinetic process, DO values slowly shift from 100 mV to 150 mV, in a similar way as the initial stage.

The Nernst equation is based on a 1:1 ratio stoichiometric relationship of the species involved in the chemical equation and can be used to relate the measured ORP values that indicate the degree of the chemical reaction.

$$\text{ORP} = -340 \text{ mV} + \left( \frac{0.059}{6} \cdot \text{Log DO} \right) + \{61 \cdot \text{pH} - 59.88 \cdot \text{Log} \frac{[NH_4]}{[NO_2]}\} \quad (12)$$

Similar results were found from empirical data analyzed by Anthonisen [51] as regards pH and FA concentration; besides, the values obtained in this work were coincident with the Pourboix graph and the thermodynamics parameters [32].

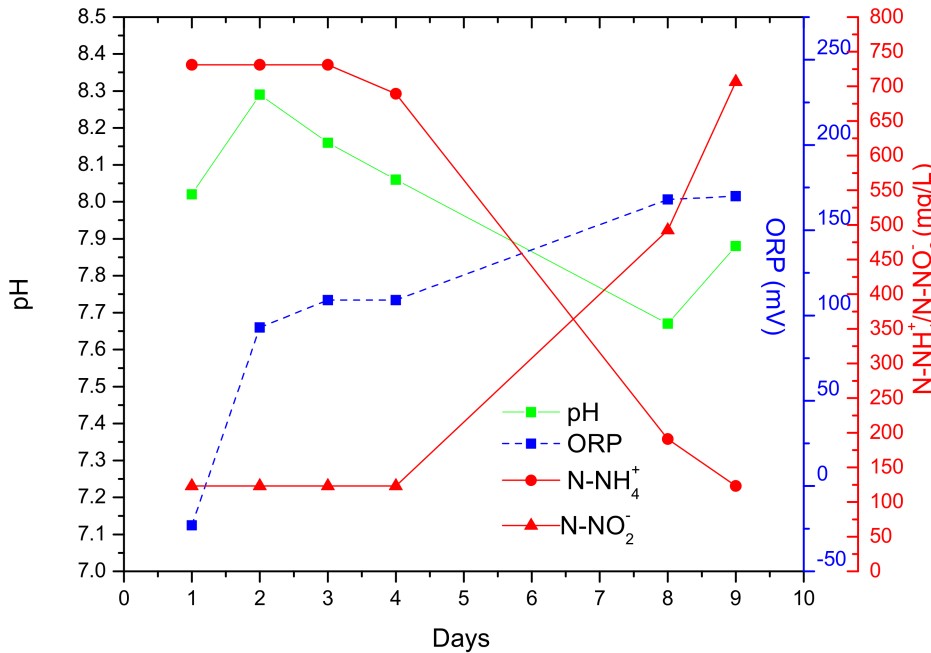

**Figure 6.** Evolution of nitrification process and the relation to ORP.

Linear non-equilibrium thermodynamics was implemented to simulate the biological nitrification process and compared it to ammonium oxidation. Empirical variation in the ORP was obtained in measuring changes in $NH_4^+$, $NO_3^-$ and $NO_2^-$ concentrations (Figure 7). While in Figure 8, it is shown the correlation between empirical and simulated ORP. The results indicate acceptable correlations on molarities of electrons transferred. The simulation model was completed based on the Nernst equations for ammonium oxidation, including a mathematical model to evaluate the influence of variables as pH, DO, and $NH_4^+$. The results can be interpreted as a correct qualitative description of this novel model concept.

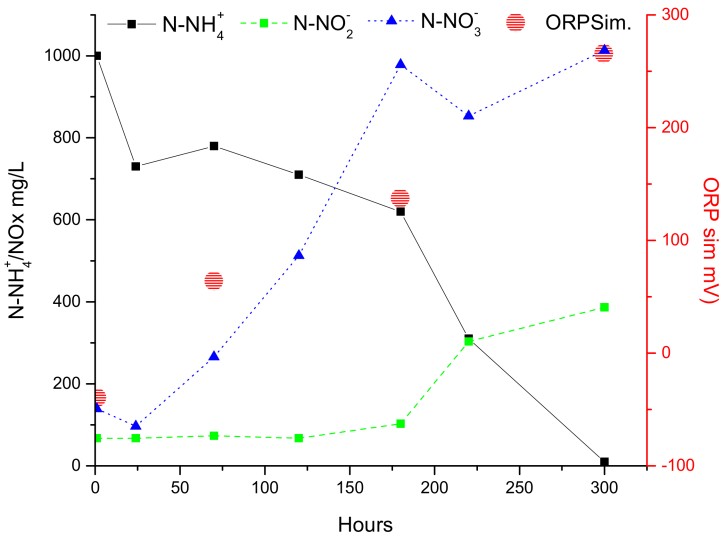

**Figure 7.** Simulation derivate by ORP and empirical data (red circle) and the evolution of nitrogen oxidation.

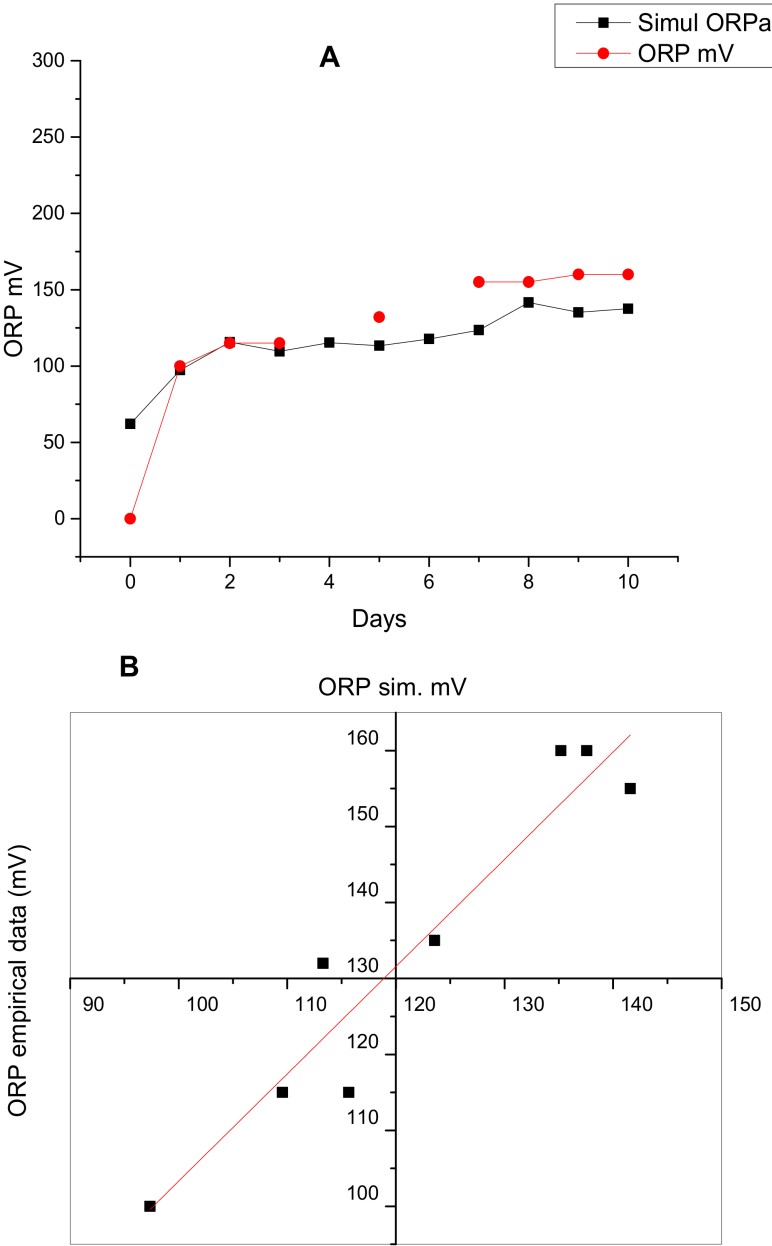

**Figure 8.** Simulated nitrification over ORP values. (**A**) values obtained by simulation and (**B**) Regression analysis between empirical and simulated data.

These findings are coincident with previous observations [36,50] where the redox reactions reflect the transfer of an electron between chemical species and bacteria (catalytic). The reaction is governed by thermodynamic parameters such as nitrogen species concentration, DO, pH, and ORP [37]. There is a three-way convergence among the thermodynamics of half-reactions (ORP), the physiology of microorganisms (DO), and the presence of chemical constituents in naturally electron acceptors, that dominate physiological reactions of microorganisms which provides criteria for monitoring nitrifying bacteria activities.

Hence, the results obtained using the ORP model where the intersection of ORP, DO and pH are represented. The best conditions for nitrifying bacteria are identified as red zones where oxidative condition prevails and the Nernst equation has the major increment in the oxidation of ammonium (Figure 9).

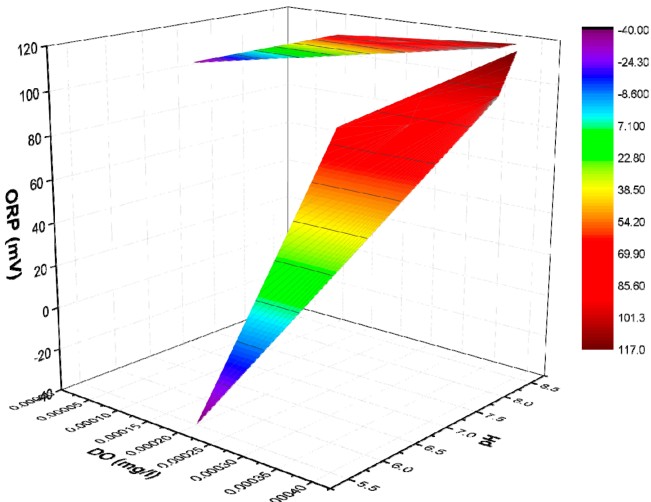

**Figure 9.** The intersection of ORP, DO, and pH indicating the best conditions for nitrifying bacteria.

*3.3. Inhibition Assays: Maximum Load of BRNW*

In general, research articles focused on operating a simulated effluent or by using simulation software [55,56]. In this work, the Monod model and Nernst equation were tested on diluted BRNW.

As expected, concentrations of BRNW above 20% inhibit ammonia oxidizers as it is shown in Figure 10 (green line) while nitrification activities were not affected at 5, 10, and 15% of BRNW dilutions, the ratio 50% showed total inhibition (data not shown). Probably, the inhibition has been induced by free ammonia due to its high concentration and pH; for example, to 850 mg/L N-NH$_4$$^+$ at 25 °C and pH = 7.8 FA = 27 mg/L; however at pH = 7.2 FA = 7.49 mg/L. Furthermore, nuclear wastewater contains uranium (600 mg/L) and TBP which presents high toxicity and inhibits bacterial growth.

Nitrate is not observed at 15% because COD concentration could favor a simultaneous denitrification process. This observation is related to Figure 10A, where N-NH$_4$$^+$ decrease is evident at that conditions. This result could be due to denitrification bacteria already present in the wastewater (Figure 10C). Nitrite were not observed in all dilutions of BRNW, probably for the complete oxidation of ammonium by bacteria.

In previous studies, it was demonstrated that attached bacteria has a better performance compared to planktonic cultures. The major advantage of employing nitrifying attached biomass is that they can provide a relatively consistent culture. Added to that, other parameters such as pH, DO and ORP can be adjusted to bring the best physicochemical environment for nitrifying attached bacteria.

The process was performed with low COD/N ratios in BRNW. A carbonate source was added to the media to increase the inorganic carbon to enhance autotrophic bacteria.

Uranium present in BRNW form complex with carbonate. Carbonate complexes are comparable to chelating agents as EDTA, NTA, and DTPA which alter treatment performance due to different equilibria. The concentrations of these species are governed by the acid dissociation constant K = [H$^+$].[HCO$_3$$^-$]/[H$_2$CO$_3$].

Once the controlled parameters assessed in previous assays were adjusted to operate in a BRNW, a longer treatment period (100 days) was set in a bioreactor (see Section 2.7). Figure 11 shows the cultures in a semi-continuous process with an agitation of 120 rpm to guarantee no limitations in the oxygen supplies. The immobilized bacteria culture showed, at the final stage of the process, an increment of the nitrification performance which is probably due to the biofilm maturity of the bacterial consortium [57,58].

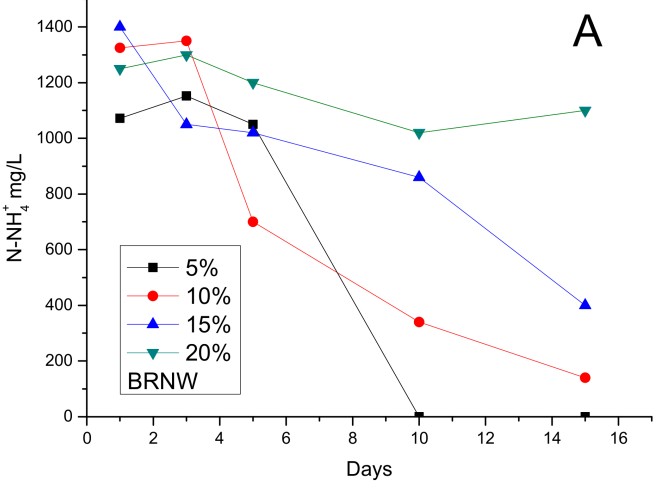

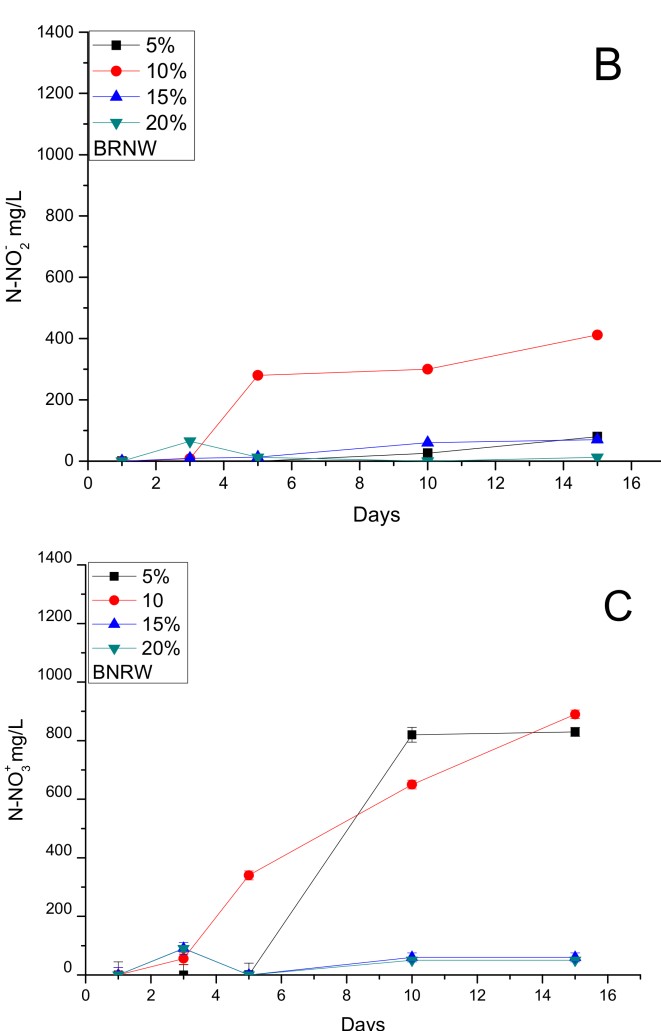

**Figure 10.** Inhibition of nitrification by BRNW. Performance of (**A**) N-NH$_4^+$mg/L, (**B**) N-NO$_2^-$ mg/L and (**C**) N-NO$_3^-$ at different BRNW concentrations.

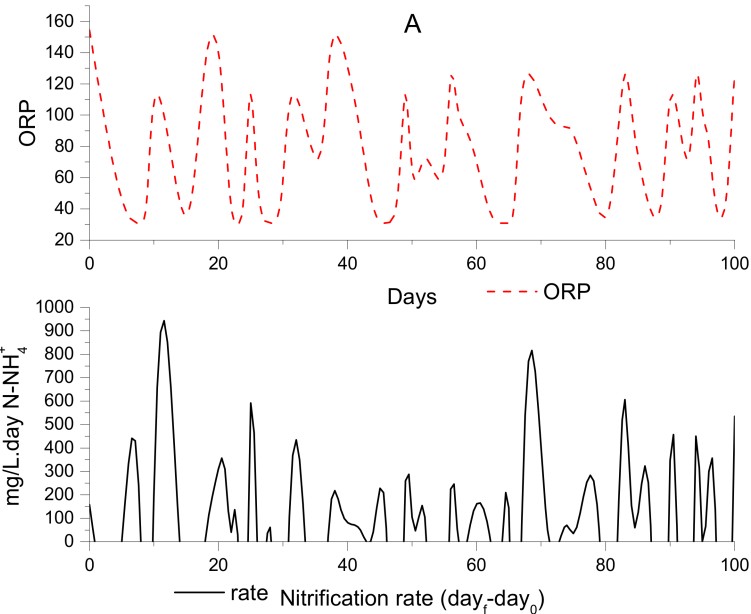

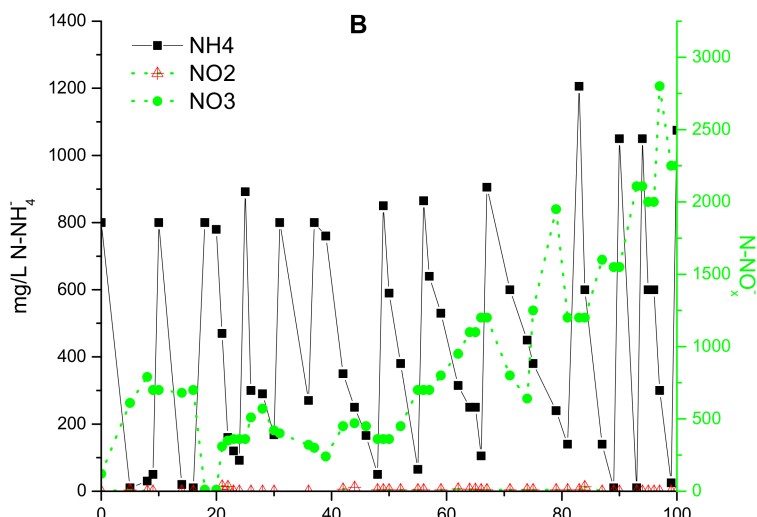

**Figure 11.** Nitrification in semi-continuous process at 100 days with attached biomass: (**A**) ORP measured (above figure) and nitrification rate (below figure) (**B**) Ammonium conversion in the nitrification process.

One of the most striking features is the performance of modelization applied on real wastewater in a large period of time. In scale-up operation conditions, the nitrification process was performed under ORP control within a range, based on the Monod model, to ensure an oxidant environment to enhance nitrifier bacteria, maintaining DO levels, pH, and temperature.

It was observed that the model can be scaled-up for the design of a pilot nitrifying bioreactor and the operation conditions were defined as follows:

$$V = 4.55 \cdot \left\{ \left( 251 \frac{mg}{l} \cdot day \right) \cdot (7.2 - pH) \cdot e^{\theta(T-Tr)} \cdot \left[ \frac{OD}{(Ks_{OD} + OD)} \right] \cdot \left( \left[ \frac{(N - NH_4^+)}{(N - NH_4^+) \cdot \left( 1 + \left( \frac{HNO2}{1.25} \right) \right)} \right] \right) \right\}_{25\,°C} \quad (13)$$

The ORP model used was according to the next expression:

$$E = -340 \text{ mV} + \left( \frac{0.059}{6} \cdot \text{og DO} \right) + \{61 \cdot pH - 59.88 \cdot \text{Log} \frac{[NH_4^+]}{[NO_x^-]} \} \tag{14}$$

These results demonstrate the importance of combining ORP and nitrification process in complex nuclear effluents, allowing a more tuned monitoring. In this case, competition between heterotrophs and AOB can be avoided by ORP and pH manipulation.

Roy [59] shows that Kaczorek and Ledakowicz obtained a high oxidation rate on real effluent. Their study was performed from a reactor treating wastewater containing 340 mg $N-NH_4^+$ l/d with a Vmax: 116–161 mg $N-NH_4^+$ l/d. However, the data reported by them shows a higher nitrification rate at low substrates and stronger inhibition at higher substrate concentrations. Several studies proved the tight relationship between ORP and biochemical pathways of nitrification, and by using the Monod model the variation of the pH, DO, and TAN could be modeled to establish an online control of the process.

## 4. Conclusions

Recent studies have suggested that there are more efficient biological alternatives to eliminate ammonia than using conventional nitrification processes. Considering that the novel processes do not require neither oxygenation nor long times to transform ammonia into nitrite or nitrate. However, such new technologies were not used for nuclear effluents where a high load of nitrogen and COD represent an unfavorable condition for microorganisms. In this work, it was found that under strict control of physicochemical parameters the activity of nitrifier bacteria could be adjusted to optimize the performance even in an effluent with a high load of nitrogen and COD.

In addition, the advantage of the present methodology consisted of using a Blended Real Nuclear Wastewater (BRNW) that helps neutralize acid pH present in the nuclear wastewater. The principal advantages of this procedure were not only reducing the cost of reagents and treating the whole effluent but also contributing to obtain acceptable oxidation of ammonia despite a high load of nitrogen and COD and a large period of effluent treatment.

Predicting parameters obtained by the Monod model were corroborated in an oxidation environment with ORP controlled in a high strength condition with real nuclear wastewater. The techniques and models used in this work show great potential for oxidizing ammonium from real complex nuclear wastewater with low cost and robust performance.

The next step to prove the efficiency of the methodology proposed involves a continuous operation at a pilot scale and continue with denitrification. This process represents sustainable biotechnology to recycling water from the uranium industry.

**Author Contributions:** Conceptualization, M.V.; methodology, M.V.; software, M.V.; validation, M.V. and P.S.P.; formal analysis, M.V., A.R. and P.S.P.; investigation, M.V., A.R. and P.S.P.; resources, M.V. and P.S.P.; data curation, M.V.; writing—original draft preparation, M.V., A.R. and P.S.P.; writing—review and editing, M.V., A.R. and P.S.P.; visualization, M.V., A.R. and P.S.P.; supervision, M.V. and P.S.P.; project administration, M.V. and P.S.P.; funding acquisition, M.V. All authors have read and agreed to the published version of the manuscript.

**Funding:** This research received no external funding from National Commission of Atomic Energy.

**Institutional Review Board Statement:** Not applicable.

**Informed Consent Statement:** Not applicable.

**Conflicts of Interest:** The authors declare no conflict of interest.

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
