# Peer review of "Nitrification Process in a Nuclear Wastewater with High Load of Nitrogen, Uranium and Organic Matter under ORP Controlled"

_water, doi:10.3390/w13111607_

Round 1
Reviewer 1 Report
The manuscript ID water-1195042 entitled „Nitrification process in a nuclear wastewater with high load of nitrogen, uranium and organic matter under ORP controlled” was under consideration. The paper presents research results and could be valuable, but in the present form, the publication is not recommended because of the low quality of description and mismatch of information. The manuscript is poorly prepared, therefore in some fragments, it was not possible to follow the Authors' thoughts.
1) lines 90-94: the text seems to be conclusions rather than aim or scope
2) The two types of industrial wastewater (processing 1 and washing 2) were mixed in 1:10 proportion. It was helpful to obtain neutralized wastewater with diluted N concentration. Please inform the readers about the potential of such a solution in practice. What are the daily amounts of these two types of wastewater? Is it realistic to use this proportion of wastewater mixing in a factory?
3) The methodology needs reediting. It is wide, but the description of the main concept of experiments is omitted. Describe better the experiments. Identify their interrelationships. Please specify dependent and independent variables. Please specify the application of blended wastewater in several experiments.
4) The symbols applied in equations are not explained in the text!!! What is the point of such writing? Need to correct it. The list of symbols will be helpful.
5) The authors do not care about the readers and reviewers. They use different symbols to describe one parameter (probably because there is no description in the text), e.g. Vmax, μmax, mmax, Vel (fig.1)
6) In methodology firstly is described the medium to inoculation (2.1.) and a few paragraphs later (2.4) the isolation is described. I propose to connect the information.
7) The wastewater is characterized (lines 109-115) and in not existed Table S1!!!!!
8) What table 1 is presented? Lines 118 and 123 present different information. Data in table 1 are not consistent with wastewater characteristics from the text.
9) Table 1 isn’t in MDPI style: purple highlight?
10) The 2.3. paragraph is lost.
11) line 130: incomplete parenthesis
12) line 131 “indophenol blue absorption spectrophotometry” rather spectrophotometric determination of ammonia ion using indophenol blue method at ….. nm
13) The Authors use the unit “mv”. Why, when the symbol of volt is “V”? Correctly is “mV”.
14) Figure is inserted twice.
15) All equations should be improved – down dots are using instead of multiplication sign, incomplete parenthesis, different parenthesis types at the same level
16) Try to reduce the size of the figures and compact the figures - make better use of space on the page. Now sometimes is one or two figures on one page, whereas on the next page is a few text lines. Little work and many benefits for the reader.
17) lines 67-68, double ref. [25]
18) line 69 – use the full name of SND, like previous acronyms
It is difficult to assess the substantive value when wading through a maze of errors and understatements. Please, prepare the entire manuscript correctly before the review. Unfortunately, the poor quality of preparation calls into question the scientific soundness.
Author Response
MDPI AG,
St. Alban-Anlage 66
4052 Basel, Switzerland
Tel.: +41 61 683 77 34
Fax: +41 61 302 89 18
Buenos Aires, April 26th of 2021
Dear Editor-in-Chief,
Please find enclosed the revised manuscript entitled “Nitrification process in a nuclear wastewater with high load of nitrogen, uranium and organic matter under ORP controlled” by Mariano Venturini as corresponding author.
We appreciated the in-depth observation and revision done by the reviewers of our manuscript. The revision allows us to improve the text, reduce some redundant information and make it clearer for the readers.
Down below, the Editor will find the responses to both the reviewers. It is in our desire that the manuscript may attend the requirements to consider it for publication.
Thanks for your consideration and I look forward to hearing from you,
Sincerely,
Dr. Mariano Venturini
1) lines 90-94: the text seems to be conclusions rather than aim or scope
Answer: We agree with the reviewer and the paragraph mentioned: “The hypothesis is that ORP is one of the most important factors to control and maintain an autotrophic population in real wastewater and for scale-up conditions” was removed from the text, to make clearer the objective of this work for “According to Chang [35], Nernst equations for on-line control of the biological process was based on a one-to-one stoichiometric relation for the oxidizing and the reducing species [30]. In the present work, ORP changes in the treatment of nuclear wastewater effluent was investigated”
2) The two types of industrial wastewater (processing 1 and washing 2) were mixed in 1:10 proportion. It was helpful to obtain neutralized wastewater with diluted N concentration. Please inform the readers about the potential of such a solution in practice. What are the daily amounts of these two types of wastewater? Is it realistic to use this proportion of wastewater mixing in a factory?
Answer: We appreciated the Review´s comments and we find that our proposal is feasible to implement a reengineering at a real nuclear treatment plant facilities saving reagents, pipelines, and reactors tanks. Besides, the text was addressed at the Conclusion part with the following texts. Lines ( 133-149) “Real nuclear wastewater samples consisted of a mixing of effluent streams from the argentine uranium conversion facility. This Blended Effluent from Real Nuclear Wastewater, named as BRNW hereafter, was composed by a mix of different discharge flows, so as to neutralize and dilute nitrogen concentration [41]. The first (1) stream was the current processing characterized by a pH=1 and a concentration of nitrogen that varies between 11000-14000 mg/l (3 N-NH4+ /1 N-NO3-). The second (2) stream came from a domestic industrial washing equipment effluent with a pH=9 and 2.7 g/l total organic nitrogen and 13000 mg/l of COD with small quantities of TriButilPhosphate (TBP) and detergent. Both effluents were mixed in a variables relation (around 1/10) to ensure that the final parameters were: COD 9000 mg O2/l, N-NO3- 4000 mg/l vs 1500 and N- NH4+ 7000 mg/l vs 1.4. The selected values of the parameters were obtained from the Monod maximum velocity (Vmáx) and ORP
determination, with the final pH adjusted to pH=7.
The table 1 reveals that there have been three different periods to make acclimation. Strategies were carried out in three stages for 40 days. The first stage was with a synthetic medium, after that, similar Nitrogen content was incorporated with quantities of COD. The final stage was a process that incorporated a BRNW (Table 1)”
3) The methodology needs reediting. It is wide, but the description of the main concept of experiments is omitted. Describe better the experiments. Identify their interrelationships. Please specify dependent and independent variables. Please specify the application of blended wastewater in several experiments.
Answer: The reviewer mentioned the need to reedit and improve the text of the experiments. The text was revised and adjusted to better describe how the experiments were performed. Lines 170 -178: 2.3 Test analysis, the text was rewritten. Lines 195-207 2.4 Monod Kinetics model, the text was rewritten
As it was solicited in Line 399, it was added what variables are dependent and what are independent
As the reviewer pointed out, ot was added and clarify when the experiment performed used the blended wastewater
4) The symbols applied in equations are not explained in the text!!! What is the point of such writing? Need to correct it. The list of symbols will be helpful.
Answer: We agree with the reviewer and the text was modified to reduce the number of equations and those that remained in the text were adequately described. All along the text the symbols were identified and correction are as “track changes”
5) The authors do not care about the readers and reviewers. They use different symbols to describe one parameter (probably because there is no description in the text), e.g. Vmax, μmax, mmax, Vel (fig.1)
Answer: We deeply apologize with the reviewer and we recognize that there were typing errors that are thoroughly addressed and corrected in the manuscript. In the text the reviewer will find the correction as “track changes”
6) In methodology firstly is described the medium to inoculation (2.1.) and a few paragraphs later (2.4) the isolation is described. I propose to connect the information.
Answer: We appreciated the comment and we proceeded accordingly and the text was unified. Lines 106-118. 2.1 Isolation Synthetic medium.
7) The wastewater is characterized (lines 109-115) and does not exist Table S1!!!!!
Answer: As the reviewer identified it was a mistake in Table S1, that corresponds to the same Table 1. It was adjusted in the manuscript. Besides, Table 1 presents the acclimatation medium used to adapt bacterial inoculum, previously isolated from soil and industrial sludge. The epigraph of the Table 1, was also modified to better represent the information that it contains. Line 149. Table 1. Characteristics of the acclimatation media for bacterial adaptation.
We introduce the following text: “it was used two different stream current to supplement the synthetic media: Stage 1, a synthetic media without nitrogen source, received a volumen from stream 1 sample up to obtain a 500-700mg/l ammonium concentration in the media. Stage 2, was a synthetic medium with a sample of stream 2, incorporating COD to produce heterotrophic acclimatation. The final Step consisted of an adaptation of the culture to BRNW reaching a pH, nitrogen content and COD to optimal growth of nitrifying bacteria”.
8) What table 1 is presented? Lines 118 and 123 present different information. Data in table 1 are not consistent with wastewater characteristics from the text.
Answer: Regarding this question, the reviewer can find the answer to this question related to the text presented in the previous point 7.
9) Table 1 isn’t in MDPI style: purple highlight?
Answer: The color in table 1 is removed as was mentioned by the reviewer.
10) The 2.3. paragraph is lost.
Answer: Unfortunately, we could not understand what the reviewer means by the missing paragraph
11) line 130: incomplete parenthesis
Answer: The mistake the reviewer mentioned was addressed and remove form the text in the manuscript
12) line 131 “indophenol blue absorption spectrophotometry” rather spectrophotometric determination of ammonia ion using indophenol blue method at ….. nm
Answer: What the reviewer suggested to replace in line 131, was addressed by the following text “Absorption spectrophotometric determination of ammonia was carried out by phenate method at 640 nm (4500-NH3 FB)”.
13) The Authors use the unit “mv”. Why, when the symbol of volt is “V”? Correctly is “mV”
Answer: The typing error was amended all along the manuscript and was modify as suggested “mV”
14) Figure is inserted twice.
Answer: The figure that was doubled was removed from the manuscript. There was another graph that corresponded to that one replicated. The change was already done for Figure 3.
15) All equations should be improved – down dots are using instead of multiplication sign, incomplete parenthesis, different parenthesis types at the same level
Answer: We appreciated the reviewer suggestions to adequate the formal indication on how to express the equations and all the equations were adjusted (from 1 to 13)
16) Try to reduce the size of the figures and compact the figures - make better use of space on the page. Now sometimes is one or two figures on one page, whereas on the next page is a few text lines. Little work and many benefits for the reader.
Answer: As solicited by the reviewer, we reduce the size of the figures, however, we did not move the figures position, to prevent the reference number alteration, and facilitate the reviewers to identify the corrections performed. After unifying the correction at the final version, we can organize the position of all the graphs.
17) lines 67-68, double ref. [25]
Answer: As the reviewer indicated the double reference was removed
18) line 69 – use the full name of SND, like previous acronyms,
Answer: As the reviewer indicated the full description of acronym SND was included (Line 69). It stands for “simultaneous nitrification-denitrification (SND)”

Reviewer 2 Report
Review of
" Nitrification process in a nuclear wastewater with high load of nitrogen, uranium and organic matter under ORP controlled "
by
Mariano Venturini, Ariana Rossen , Patricia Silva Paulo
The results presented in the manuscript are interesting and worth publishing. The main disadvantage is the numerous editorial errors that make it difficult to properly evaluate the results.
The manuscript needs to be revised in order to publish.
General comments:
- the authors of the manuscript very often introduce records of the measured or determined parameters without explaining them (e.g. what does NO3K mean in Fig. 7) - this requires ordering the records and adding explanations throughout the manuscript
- the manuscript presents descriptions for different temperatures (usually 20 or 25 degrees C) - please explain why
- Introduction: with reference to sentence “Although this process is difficult to implement in an effluent with a high load of COD …..” the possibility of using this technology for wastewater treatment from nuclear plants should be presented, together with a reference to publications on this subject (e.g. Joanna Majtacz, Dominika Grubba and Krzysztof Czerwionka. Application of the Anammox Process for Treatment of Liquid Phase Digestate. Water 2020, 12, 2965; doi:10.3390/w12112965)
- 1 Synthetic medium: sentence “Inoculum was initialized at 5% v/v at 25° C - 120rpm in 500ml Erlenmeyer flask with a final volume of 250ml” requires an explanation (what inoculum, why at 25 degrees C, why only basic synthetic medium without ammonium nitrogen was used)
- 2 Blended real nuclear wastewater (BRNW): table 1 (or maybe S1?) is completely unreadable - it should be corrected by completing the descriptions for the 3 test periods. Why is the COD concentration shown in different units (mg / l or g / l or % v / v)?
- 4 Nitrifying bacteria: with reference to the sentence “Therefore, nitrifying biomass was isolated from a soil sample in synthetic medium, and then acclimated to an organic load.” it is necessary to explain: 1) why biomass from sewage treatment plant was not used 2) what was the idea of acclimatization of nitrifying bacteria to organic load.
- 5 Monod Kinetics model: this chapter requires a complete conversion (there are references to equations that do not exist, the definition (and notation) of the u parameter is incorrect, it contains many formal errors, and requires a thorough explanation of the method of determining equation parameters)
- 6 Control of environmental …: with reference to the sentence “The original equation proposed by Chang [35] was modified by replacing ln(1/[H+]) for 2.3026 × 4 ? (??), being as follow” it is necessary to explain: 1) to which equation does this entry refer 2) on what basis the modification was made
- 2: 1) the vertical axis shows the concentration of ammoniacal nitrogen - therefore there is no additional vertical axis describing the nitrification rate; 2) the description of the horizontal axis is incomprehensible
- equation 10: a detailed description of the parameters presented in equation 10 should be provided (please also verify the correctness of their notation) - a comparison of this form of equation with the literature should also be presented
- Point 3.2: The description should be supplemented with a comparison of the obtained results in relation to the values for the nitrification process for typical municipal wastewater
- Point 3.3: with reference to the sentence “Probably the inhibition has been induced by free ammonia due to its high concentration (850 mg/l N-NH4+).” 1) 850 mg NH4-N is not a very high concentration - much higher values (exceeding 2000 mg / l) occur in the effluents from digestate dewatering from fermentation chambers, and they are effectively purified in the deammonification process; 2) the presence of free ammonia is a function of pH - if the authors expect the presence of free ammonia, this should be confirmed based on calculations based on the pH value
- Conclusions: at this point, the most important achievements of the presented research should be presented - they are too extensive in the current version. This point should be edited again paying special attention to the novelty of the presented research.
Detailed comments (DC - the places marked in the manuscript):
- the meaning of the abbreviation IAEA should be presented
- the meaning of the abbreviation ZLD should be presented
- re-reference to item 25
- this entry needs to be corrected - it is illegible
- the meanings of all the coefficients are not explained and some are explained incorrectly
- this sentence is repeated
- the abbreviation is not explained and the equation is illegible (it is rather a record of interdependencies)
- writing this equation requires improvement and clarification of the components
- this information has already been presented in point "2.3 Test analysis"
- it is not a load but a concentration
- I guess nitrite acid?
- this is not the correct figure - fig. 4 was repeated here
- the description of the vertical axis is illegible - 1) whether it is a% of saturation; 2)% in relation to the assumed value 3) if DO is in% then the unit cannot be mg / l
- this is a misspelling
- rather DO
- there are three graphs in Fig 11 and only two of them are described
- ???
- this abbreviation was previously introduced

Author Response
MDPI AG,
St. Alban-Anlage 66
4052 Basel, Switzerland
Tel.: +41 61 683 77 34
Fax: +41 61 302 89 18
Buenos Aires, April 27th of 2021
Dear Editor-in-Chief,
Please find enclosed the revised manuscript entitled “Nitrification process in a nuclear wastewater with high load of nitrogen, uranium and organic matter under ORP controlled” by Mariano Venturini as corresponding author.
We appreciated the in-depth observation and revision done by the reviewers of our manuscript. The revision allows us to improve the text, reduce some redundant information and make it clearer for the readers.
Down below, the Editor will find the responses to both the reviewers. It is in our desire that the manuscript may attend the requirements to consider it for publication.
Thanks for your consideration and I look forward to hearing from you,
Sincerely,
Dr. Mariano Venturini
MDPI AG,
St. Alban-Anlage 66
4052 Basel, Switzerland
Tel.: +41 61 683 77 34
Fax: +41 61 302 89 18
Buenos Aires, April 27th of 2021
Dear Editor-in-Chief,
Please find enclosed the revised manuscript entitled “Nitrification process in a nuclear wastewater with high load of nitrogen, uranium and organic matter under ORP controlled” by Mariano Venturini as corresponding author.
We appreciated the in-depth observation and revision done by the reviewers of our manuscript. The revision allows us to improve the text, reduce some redundant information and make it clearer for the readers.
Down below, the Editor will find the responses to both the reviewers. It is in our desire that the manuscript may attend the requirements to consider it for publication.
Thanks for your consideration and I look forward to hearing from you,
Sincerely,
Dr. Mariano Venturini
Revisions from Reviewer 2
General comments:
- The authors of the manuscript very often introduce records of the measured or determined parameters without explaining them (e.g. what does NO3K mean in Fig. 7) - this requires ordering the records and adding explanations throughout the manuscript
Answer: The reviewer mentioned that there must be more explanations of the parameters measured and /or used for each experiment so we proceeded accordingly and introduced a text each time that was required so as to make that measurement clearer. Revise Line 484
- The manuscript presents descriptions for different temperatures (usually 20 or 25 degrees C) - please explain why
Answer: It was corrected and the temperature used all along was 25 °C. In the case of 20°C was a typing error.
- Introduction: with reference to sentence “Although this process is difficult to implement in an effluent with a high load of COD …..” the possibility of using this technology for wastewater treatment from nuclear plants should be presented, together with a reference to publications on this subject (e.g. Joanna Majtacz, Dominika Grubba and Krzysztof Czerwionka. Application of the Anammox Process for Treatment of Liquid Phase Digestate. Water 2020, 12, 2965; doi:10.3390/w12112965)
Answer: We thank the reviewer for the reference, which was included in the bibliography. In Line 81 is mentioned in this reference, identity as [31]. Besides, we introduce a text in conclusions that reinforce the idea that this biotechnology using nitrifier bacteria to improve the nitrogen removal from a nuclear effluent seems to be a promising alternative.
- 1- Synthetic medium: sentence “Inoculum was initialized at 5% v/v at 25° C - 120rpm in 500ml Erlenmeyer flask with a final volume of 250ml” requires an explanation (what inoculum, why at 25 degrees C, why only basic synthetic medium without ammonium nitrogen was used)
Answer: In this case, the text in Paragraph between the Lines 110 to 115 was corrected and rewrite. First, the temperature was adjusted to 25 °C as an optimal temperature for bacterial metabolism. The used inoculum corresponds to the bacteria isolated from soil and nuclear industrial sludge.
- 2 Blended real nuclear wastewater (BRNW): table 1 (or maybe S1?) is completely unreadable - it should be corrected by completing the descriptions for the 3 test periods. Why is the COD concentration shown in different units (mg / l or g / l or % v / v)?
Answer: Absolutely right, it was a typing error that was revised and adjusted in the new manuscript. Table S1 corresponds to Table 1. Related to COD has its unit in mg/l. “ The table 1 reveals that there have been three different periods to make acclimation. Strategies were carried out in three stages for 40 days. The first stage was with a synthetic medium, after that, similar Nitrogen content was incorporated with quantities of COD. The final stage was a process that incorporated a BRNW (Table 1)”
Lines (133-149) “Real nuclear wastewater samples consisted of a mixing of effluent streams from the argentine uranium conversion facility. This Blended Effluent from Real Nuclear Wastewater, named as BRNW hereafter, was composed by a mix of different discharge flows, so as to neutralize and dilute nitrogen concentration [41]. The first (1) stream was the current processing characterized by a pH=1 and a concentration of nitrogen that varies between 11000-14000 mg/l (3 N-NH4+ /1 N-NO3-). The second (2) stream came from a domestic industrial washing equipment effluent with a pH=9 and 2.7 g/l total organic nitrogen and 13000 mg/l of COD with small quantities of TriButilPhosphate (TBP) and detergent. Both effluents were mixed in a variables relation (around 1/10) to ensure that the final parameters were: COD 9000 mg O2/l, N-NO3- 4000 mg/l vs 1500 and N- NH4+ 7000 mg/l vs 1.4. The selected values of the parameters were obtained from the Monod maximum velocity (Vmáx) and ORP”
The epigraph of the Table 1, was also modified to better represent the information that it contains. Line 149. Table 1. Characteristics of the acclimatation media for bacterial adaptation.
4 Nitrifying bacteria: with reference to the sentence “Therefore, nitrifying biomass was isolated from a soil sample in synthetic medium, and then acclimated to an organic load.” it is necessary to explain: 1) why biomass from sewage treatment plant was not used 2) what was the idea of acclimatization of nitrifying bacteria to organic load.
Answer: Absolutely right, it was a typing error that was revised and adjusted in the new manuscript. we proceeded accordingly and the text was unified. Lines 112-124. 2.1 Isolation Synthetic medium
- 5 Monod Kinetics model: this chapter requires a complete conversion (there are references to equations that do not exist, the definition (and notation) of the u parameter is incorrect, it contains many formal errors, and requires a thorough explanation of the method of determining equation parameters)
Answer: Absolutely right, it was a typing error that was revised and adjusted. The text was revised and adjusted to better describe how the experiments were performedLines 195-207 2.4 Monod Kinetics model, the text was rewritten. Line 201-278, we were including reference of expressions, the Operational factor in line 242.
- 6 Control of environmental …: with reference to the sentence “The original equation proposed by Chang [35] was modified by replacing ln(1/[H+]) for 2.3026 × 4 ? (??), being as follow” it is necessary to explain: 1) to which equation does this entry refer 2) on what basis the modification was made
Answer: Absolutely right, it was a typing error that was revised and adjusted in the new manuscript. The reference of the equations were incorporate in line 285 and the reference for all expressions. that there were typing errors that are thoroughly addressed and corrected in the manuscript. In the text the reviewer will find the correction as “track changes”
- 2: 1) the vertical axis shows the concentration of ammoniacal nitrogen - therefore there is no additional vertical axis describing the nitrification rate; 2) the description of the horizontal axis is incomprehensibleFIG 2
Answer: Absolutely right, it was a typing error that was revised and adjusted in the new graph and legend
- equation 10: a detailed description of the parameters presented in equation 10 should be provided (please also verify the correctness of their notation) - a comparison of this form of equation with the literature should also be presented
Answer: Absolutely right, it was a typing error that was revised and adjusted in the new manuscript. Independent and variables were described in line455-457 and the legend was incorporated in line 465-466.
- Point 3.2: The description should be supplemented with a comparison of the obtained results in relation to the values for the nitrification process for typical municipal wastewater.
Answer: We appreciated the comment of the reviewer, however, as we intend to optimize the nitrification on wastewater from the process to transform the uranium into nuclear fuel, the blended type of effluents is not conventional wastewater due to the different components and uranium concentration (considered as a radioactive effluent) and originally, we did not consider it worth comparing both types of wastewater effluents nitrification processes.
- Point 3.3: with reference to the sentence “Probably the inhibition has been induced by free ammonia due to its high concentration (850 mg/l N-NH4+).” 1) 850 mg NH4-N is not a very high concentration - much higher values (exceeding 2000 mg / l) occur in the effluents from digestate dewatering from fermentation chambers, and they are effectively purified in the deammonification process; 2) the presence of free ammonia is a function of pH - if the authors expect the presence of free ammonia, this should be confirmed based on calculations based on the pH value
Answer: We appreciated the comment of the reviewer, the FA values were included in line 569-570 as pH function according to equation (5). Effectively, 850 mg/l of NH4+is not a very high concentration however, in the nuclear effluent with uranium and COD, makes us think that it is significant in this context. a study should be done to establish these parameters but it is not the objective of this work
- Conclusions: at this point, the most important achievements of the presented research should be presented - they are too extensive in the current version. This point should be edited again paying special attention to the novelty of the presented research.
Answer: Absolutely right the line 683-693 were removed from the conclusion.
- Detailed comments (DC - the places marked in the manuscript):
We appreciate the reviewers comments:
- the meaning of the abbreviation IAEA should be presented.
Addressed: Line 42 International Atomic Energy Agency (IAEA)
- the meaning of the abbreviation ZLD should be presented
Addressed: Line 59 Zero Liquid Discharge (ZLD)
- re-reference to item 25. this entry needs to be corrected - it is illegible
Addressed: re-reference to item 25 was modified. Line
- the meanings of all the coefficients are not explained and some are explained incorrectly
Answer: comment addressed: the coefficients and the equations were rewrite and explained each component. the correction are identified as “track changes”
- this sentence is repeated
Answer: comment addressed. The repeated sentence was removed
- the abbreviation is not explained and the equation is illegible (it is rather a record of interdependencies)
Answer: We appreciated the reviewer suggestions to adequate the formal indication on how to express the equations and all the equations were adjusted (from 1 to 13). comment addressed. It was included an abbreviation and the equation was rewritten.
- writing this equation requires improvement and clarification of the components
Answer: comment addressed. All the components of the equation are described down below it in the text
- this information has already been presented in point "2.3 Test analysis"
Answer: comment addressed. The text was rewritten to and the reduced in the explanation was removed
- it is not a load but a concentration
Answer: The reviewer is correct and corresponds to a concentration. The text was adjusted
- I guess nitrite acid?
Answer: as ammonia (uncharged), the nitrous acid (uncharge) could penetrate the cell.
- this is not the correct figure - fig. 4 was repeated here the description of the vertical axis is illegible - 1) whether it is a% of saturation; 2)% in relation to the assumed value 3) if DO is in% then the unit cannot be mg / l
Answer: We appreciate the reviewer's comment, it is correct and Fig 4 was adjusted considering the observations. Fig 4 in Line 351
- this is a misspelling rather DO there are three graphs in Fig 11 and only two of them are described ???
Answer: We thanks the reviewer for mentioning this, we proceeded accordingly and include a description of figure 11
- this abbreviation was previously introduced
Answer: As indicated by the reviewer, we removed the texts and only left the abbreviation.

Round 2
Reviewer 1 Report
Thank you very much for implementation the part of my comments in the re-submitted manuscript. In my opinion, many aspects weren’t taken into account:
- my comment No. 1. We didn’t understand each other. The text that sounds like conclusions is: “The integral equations improve on-line control of the biological process and proved to be capable of adapting the set-points related to the influent volume and the input load of nitrogen, that can be properly managed only for limited variations. Effluent treatment can be significantly improved through the application of this model and has shown to be a robust process for real wastewater samples with a high load of nitrogen and COD for nuclear effluents.” Of course, in the introduction, you hypothesize.
- my comment No. 2. I asked for clarification of the question for the readers. What volumes of type 1 and type 2 wastewater are generated in the plant per day? This may increase the credibility of such a solution based on blending 1/10. The answers are not to the question.
- my comment No. 12. Lines 157, 158: something went wrong with sentence implementation, now the incorrect word “Spectrophotometricy “ is present
The new comments:
A) Line 136: What does mean “vs 1500” and “vs 1.4”?
B) I see the lack of accuracy in presenting data concerning the BRNW/autotrophic medium ratio: line 262: 5-50%, figure 10 and text: 5-20%
C) lines 468, 469: “While nitrification activities were not affected at 5, 10 and 15% of BRNW dilutions”. On what basis do you conclude that 15% does not inhibit? Results presented in fig. 10 indicate that ammonia is removed, however, nitrite and nitrate aren’t increased. This needs to be discussed in the text.
D) fig. 5. OY axis description is “Na2CO3 ml 1M added”. It seems to be the cumulative curve, so the description “sum of Na2CO3 ml 1M added” could be more appropriate.
E) Why is the dissociation constant of carbonic acid wrongly written?
F) lines 498-500 I have a problem with these sentences because these are imprecise. In the experiment, pH was 7.7-8.3. In this range, the presence of all forms of carbonic acid is possible: CO2 (affiliated) being in equilibrium with alkalinity (domination of HCO3- and a minimal amount of CO32-). Here is a mental shortcut and it is hardly known what the Authors wanted to convey.
G) Pre scale-up operations in moving bed bioreactor (MBBR) were investigated. The bioreactor design should be presented as well as its working mode.
H) I see problems with reliability again:
- Line 131: 11-14 kg N/L ??? it is impossible; I suggest unifying the units, mg or g, not mg and g
- The resubmitted version contains the speech balloons of internal authors’ comments, I don't know how to take it?
- Line 118 – the subscript is missed
- paragraph 2.4. is omitted
- figure 12 – unit “mv” is still present
- line 137 – the Spanish letter á is present (maybe more in text)
- line 410 – rather “nitrification”
- line 471 – comma instead of dot
- equation 5 – something wrong with parenthesis
- equation 6 – something wrong with parenthesis
- equation 6 and line 226: I strongly recommend writing NH4+, not NH4-,
as well as CO32- instead of CO3-2 (and in the text) (I am concerned about reliability when I see such chemical deficiencies)
- figure 9 – no units and axis description low visible
maybe more, the Authors should do this work themselves and carefully clean the manuscript
Author Response
REVISION ADDRESSED FROM REVIEWER 1
The authors appreciated the reviewer´s time and dedication to revise the text, it helped to improve and make it clearer and concise for the readers. As it was recommended, we made a thoughtful last revision and we find other mistakes that we incorporate and underline at the end of the revision comments that go as follows.
Thank you very much for implementation the part of my comments in the re-submitted manuscript. In my opinion, many aspects weren’t taken into account:
- my comment No. 1. We didn’t understand each other. The text that sounds like conclusions is: “The integral equations improve on-line control of the biological process and proved to be capable of adapting the set-points related to the influent volume and the input load of nitrogen, that can be properly managed only for limited variations. Effluent treatment can be significantly improved through the application of this model and has shown to be a robust process for real wastewater samples with a high load of nitrogen and COD for nuclear effluents.” Of course, in the introduction, you hypothesize.
Answer: We understand each other perfectly, we follow suit. We have included that text in conclusion because we agree that that sentence summarizes our main findings after performing all the experiments and tests, and we strongly support the use of biotechnological techniques to treat nuclear wastewater containing high load of nitrogen.
- my comment No. 2. I asked for clarification of the question for the readers. What volumes of type 1 and type 2 wastewater are generated in the plant per day? This may increase the credibility of such a solution based on blending 1/10. The answers are not to the question.
Answer: Here, maybe the answer we provided was not clear in the previous revision. The facility production is not a continuous process, is a batch production according to demand. So, the effluent of line 1 and line 2 are not constant in time. The criteria for blending both effluents was to reach a neutralization point as well as the nitrogen load needed to ensure microbial nitrification, so it was an experimental stage to mix both effluents, and we estimate that it is possible this relationship, around 1/10, because domestic stream current is higher in volume than the production current.
- my comment No. 12. Lines 157, 158: something went wrong with sentence implementation, now the incorrect word “Spectrophotometricy “ is present
Answer: We thank the comment, and made the adequate correction. We introduce the correct word. “Spectrophotometric”
The new comments:
- Line 136: What does mean “vs 1500” and “vs 1.4”?
Answer: We thank the comment we have changed the texts and adjusted the numbers and the idea was to mention a “(+/-)” relationship not a “vs”. We meant that in that text each parameter value had an average. In the text, we adjusted that idea of a range in each value.
- I see the lack of accuracy in presenting data concerning the BRNW/autotrophic medium ratio: line 262: 5-50%, figure 10 and text: 5-20%
Answer: We have tried the relation of 50% that exhibit a complete inhibition of the nitrification, that is the reason why we did not include that result in the figure, however, as we have not mentioned that in so much words in the text, we include the expression “data not shown” after a brief clarification if that results.
- lines 468, 469: “While nitrification activities were not affected at 5, 10 and 15% of BRNW dilutions”. On what basis do you conclude that 15% does not inhibit? Results presented in fig. 10 indicate that ammonia is removed, however, nitrite and nitrate aren’t increased. This needs to be discussed in the text.
Answer: To better explain this conclusion to the reviewer, we would like to mention that above 15% of the BRNW dilutions The text was modified as follow to better explain this point: Nitrate is not observed at 15% because COD concentration could favor a simultaneous denitrification process. This observation is related to figure 10 A, where N-NH4+ decrease is evident at that conditions. This result could be due to denitrification bacteria already present in the wastewater (fig. 10 C)
- 5. OY axis description is “Na2CO3 ml 1M added”. It seems to be the cumulative curve, so the description “sum of Na2CO3 ml 1M added” could be more appropriate.
Answer: We have addressed that comment and we include the idea of addition of Na2CO3, so the word “sum” was incorporated into the description
- Why is the dissociation constant of carbonic acid wrongly written?
Answer: The reviewer is correct, and we have addressed that comment and we modified the dissociation constant
- lines 498-500 I have a problem with these sentences because these are imprecise. In the experiment, pH was 7.7-8.3. In this range, the presence of all forms of carbonic acid is possible: CO2 (affiliated) being in equilibrium with alkalinity (domination of HCO3- and a minimal amount of CO32-). Here is a mental shortcut and it is hardly known what the Authors wanted to convey.
Answer: Unfortunately, the mentioned paragraph was not found at the lines 489-500, so we tried to find the correspondent paragraph elsewhere in text. At line 521-524, there was a comment regarding the alkalinity equilibrium that as the reviewer mentioned is confusing and do not clarify any remarkable aspect, and we removed it (lines 509-512).
- Pre-scale-up operations in moving bed bioreactor (MBBR) were investigated. The bioreactor design should be presented as well as its working mode.
Answer: Regarding the mentioned required explanation we included a more detailed text added information about the working mode. The text goes as follow: Biomass added onto carriers were cultivated in autotrophic media for 40 days in Sartorius Biostat APlus Bioreactor. After that period, the culture was evaluated on uranium concentration. Finally, acclimated to BRNW for another 40 days with effluent up to 20%. The ORP value was kept above 100mV to guarantee the nitrification process. To establish the best nitrifier performance and avoid nitrite accumulations, the bioassay was carried out under two different values of pH, 7.2 and 7.8. Obtaining better results at 7.8. The oxygen supplied with a constant flow at 2 lpm of air and 200 rpm.
- I see problems with reliability again:
- - Line 131: 11-14 kg N/L ??? it is impossible; I suggest unifying the units, mg or g, not mg and g
Answer: The changes the review mentioned was addressed
- - The resubmitted version contains the speech balloons of internal authors’ comments, I don't know how to take it?
Answer: The balloons were removed as solicited
- - Line 118 – the subscript is missed
Answer: Thank to the reviewer´s indication we could find the mistake in the subscript of ammonia molybdite molecular formula: Na2MoO4
- - paragraph 2.4. is omitted
Answer: we accommodate the paragraph numbers. We appreciate the reviewer´s mention
- - figure 12 – unit “mv” is still present
Answer: The change was addressed.
- - line 137 – the Spanish letter á is present (maybe more in text)
Answer: The change was addressed. (máx. The correction was changed at line 144)
- - line 410 – rather “nitrification”
Answer: Although we did not find a wrong word identify as “nitrification” we revise all the words in the text
- - line 471 – comma instead of dot
Answer: comment addressed
- - equation 5 – something wrong with parenthesis
Answer: comment addressed.
- - equation 6 – something wrong with parenthesis
Answer: comment addressed and accommodate the parenthesis
- - equation 6 and line 226: I strongly recommend writing NH4+, not NH4-,
as well as CO32-instead of CO3-2 (and in the text) (I am concerned about reliability when I see such chemical deficiencies)
Answer: comment addressed and we changed the upper and subscript in ammonia and carbonate
- - figure 9 – no units and axis description low visible
Answer: comment addressed, we use a higher size of the words
- maybe more, the Authors should do this work themselves and carefully clean the manuscript
Answer: We really appreciated the reviewer´s time and dedication to revise the text, it helps to improve and make it clearer and concise. As it was recommended, we made a thoughtful last revision and we find many other mistakes, including redundant texts and results (we removed Figure 12) to reduce the text as it also was suggested by the reviewers and gain clarity. Even, we have shortened the conclusion part, making more resolutive paragraphs
The last version of the manuscript is restructured and much revised. We believe that we manage to highlight the relevance of using a controlled nitrification process using modelized parameter to remove ammonia from nuclear wastewater as a valuable cost-effective alternative technology.
Respectfully,
Mariano Venturini

Reviewer 2 Report
comments in the attached file

Author Response
Review of revised manuscript
" Nitrification process in a nuclear wastewater with high load of nitrogen, uranium and organic matter under ORP controlled "
by
Mariano Venturini, Ariana Rossen , Patricia Silva Paulo
General comments: The authors of the manuscript have substantively addressed the comments submitted and provided acceptable explanations.
The article may be published taking into account detailed comments:
- Line 156: organic nitrogen concentration should be expressed in mg/l (2,700 mg/l) similar to COD - be aware of the decimal point separating the values in thousands (i.e. 13,000 mg/l for COD)
- 5 Monod Kinetics model and 3.1 Monod kinetic parameters: all parameters in the formulas should have units defined - in the description after "where: ...."
- Fig 4: Fig 4 contains 2 graphs - an explanation of what they refer to in the title of Fig
- 10: title should be corrected “Inhibition of nitrification by BRNW monitoring A) NH4+, B) NO2- and C) NO3- concentration”, the letters A-C should also be added to the individual graphs
Addressed comments
- Line 156: organic nitrogen concentration should be expressed in mg/l (2,700 mg/l) similar to COD - be aware of the decimal point separating the values in thousands (i.e. 13,000 mg/l for COD)
Answer: We have addressed the comment and we corrected the units in the numbers; besides, we adjusted the rest of the numbers that go above thousand
- 5 Monod Kinetics model and 3.1 Monod kinetic parameters: all parameters in the formulas should have units defined - in the description after "where: ...."
Answer: We thank the reviewer to mentioned that, and we have addressed the comment and we included the units in each parameter. In the point 2.5 and 3.1.
- Fig 4: Fig 4 contains 2 graphs - an explanation of what they refer to in the title of Fig
Answer: Regarding Fig 4. We thought that we have already removed one of the figures in Fig 4. but we did not corrected the title of the fig, we thank the reviewer to spot that mistake. We have made the necessary changes. The title of Figure 4 goas as “Linearization of respirometric kinetics”
- 10: title should be corrected “Inhibition of nitrification by BRNW monitoring A) NH4+, B) NO2- and C) NO3- concentration”, the letters A-C should also be added to the individual graphs
Answer: Regarding Fig 10, we have included the number of each nitrogen parameter and in each figure, as mentioned by the reviewer.
The last version of the manuscript is restructured and much revised. We believe that we manage to highlight the relevance of using a controlled nitrification process using modelized parameters to remove ammonia from nuclear wastewater as a valuable cost-effective alternative technology. We invite the reviewer 2, go through the conclusion part, where we have shortened it making more resolutive paragraphs.
We really appreciate the comments and time for revision of the reviewer 2, and we thank the recommendation for publishing our manuscript. Hoping we have achieved the quality and level to be considered for its publication in WATER journal.
Respectfully,
Mariano Venturini
